# Male Victims of Rape: An Observational Study over Four Years in Paris, France

**DOI:** 10.3390/ijerph192113909

**Published:** 2022-10-26

**Authors:** Marc Liautard, Céline Deguette, Elizabeth Alcaraz, Hélène Diot, Patricia Vasseur, Charlotte Gorgiard, Laurène Dufayet

**Affiliations:** 1Unité Médico-Judiciaire, Hôtel-Dieu, Assistance Publique des Hôpitaux de Paris (APHP), 75004 Paris, France; 2Institut Médico-Légal de Paris, 75012 Paris, France; 3Centre Antipoison de Paris—Fédération de Toxicologie (FeTox), Hôpital Fernand-Widal, Assistance Publique des Hôpitaux de Paris (APHP), 75010 Paris, France; 4INSERM, UMRS-1144, Faculté de Pharmacie, 75006 Paris, France; 5UFR de Médecine, Université de Paris Cité, 75006 Paris, France

**Keywords:** sexual violence, rape, men, alcohol, substance use, cannabis, vulnerability

## Abstract

Sexual violence is a common and under-reported form of violence that affects all categories of individuals. We sought to provide epidemiological data on men aged 15 years and over, victims of rape or suspecting a rape, and who reported it to the police. We conducted a retrospective study at the Department of Forensic Medicine of Hôtel-Dieu, Paris, France, between 2018 and 2021. Two-hundred men were included in the study, with an average age of 28.8 years. A vulnerability was mentioned for 17.5% of them. Most of the patients reported anal penetration, committed by a single male assailant, whom they met on the day of the assault. More than 60% of the patients reported the voluntary consumption of alcohol and/or illicit substances prior to the assault. Most patients were examined shortly after the assault (median 1 day). Anal lesions were found on examination in 37.0% of patients reporting anal penetration regardless of the time frame. The presence of anal lesions was statistically higher when patients were examined within 48 h. Our results reinforce the data in the literature on the risk factors associated with sexual violence among men, notably the consumption of alcohol and illicit substances, and psychological, economic, and social vulnerability.

## 1. Introduction

Sexual violence is a common and under-reported form of violence that affects all categories of individuals, even if it is perceived and associated mostly with violence against women [1]. Worldwide victimization surveys find lower reported sexual violence among men than among women, and women usually represent more than 80% of the victims [2,3,4]. Although these numbers show a predominance of violence against women, a significant proportion of men are also victims. Awareness of sexual violence against men is quite recent and dates back to the 1980s, thanks to studies in prisons or armed conflicts [5,6]. Most experts believe that sexual violence in males is even more under-reported than in women, as men may experience stronger prejudices surrounding their sexuality [1]. Over the last years, sexual violence has been associated with psychoactive substances, in the context of drug-facilitated sexual assault (DFSA) [7]. DFSA can be “proactive”: the victim is administered, covertly or by force, a psychoactive substance; or “opportunistic”: the victim is unable to consent or to react, because of voluntary consumption of a psychoactive substance [7]. Socially, this phenomenon is associated with young women, but a recent study showed a high proportion of opportunistic DFSA in male victims of sexual assault during chem-sex [7,8]. We sought to provide epidemiological data on men aged 15 years and over, victims of rape or suspecting a rape, and who reported it to the police in Paris, France. We wanted to highlight the aggravating circumstances, including the victim’s vulnerability, and to provide epidemiologic data on adolescent and adult male victims of rape in the context of proactive or opportunistic DFSA.

## 2. Materials and Methods

### 2.1. Presentation of the Department of Forensic Medicine and of a Forensic Consultation

This Department of Forensic Medicine is the referral center for all victims of violence who reported it to the police in Paris, France, an area with more than 2.1 million inhabitants. The forensic consultation is performed only upon judicial request. In case of sexual violence, it is carried out by a forensic physician and a trained nurse. Before the examination, they explain the consultation to the patient (how it will be carried out and why) and answer any questions he might have. They also ask the victims details about their health, the assault, and the assailant; data are declarative. The consultation includes a somatic and an anogenital examination. It will allow the identification of possible traumatic lesions and the collection of judicial and medical samples. Following the consultation, the victim is systematically offered a psychological consultation and at least one medical follow-up appointment. For each initial consultation, the physician creates a computerized file, using a common framework. The judicial samples are kept in the Department and can later be sealed by the judicial authority and sent for analysis.

### 2.2. Study Methodology

We conducted a single-center retrospective study at the Department of Forensic Medicine of Hôtel-Dieu, Paris, France. We included all men aged 15 years and older at the time of the event, who consulted for rape or suspected rape between 1 January 2018 and 31 December 2021. Each file was reviewed by two forensic physicians and a nurse. Data related to the victim (age, vulnerability, previous medication, alcohol intake and its quantification: a standard drink refers to 10 g pure alcohol [9], and psychoactive substances intake), the assailant, the assault, the existence of aggravating circumstances, and data from the forensic examination were collected. Under French law, a distinction is made between rape and sexual assault. Rape is defined as any sexual penetration or oral–genital act (oral sex), of any kind, committed on the victim or by the victim on the perpetrator, by violence, constraint, threat, or surprise [10]. A sexual attack without penetration or oral sex, perpetrated under the same circumstances, is defined as a sexual assault [11]. Vulnerability can be physical, psychological, social, economic, or linked to the consumption of illicit substances or alcohol [12]. Females, males reporting sexual violence before the age of 15, and males reporting sexual violence without rape were excluded. The follow-up consultation includes only medical data (tolerance to the HIV post-exposure prophylaxis, psychological evolution, etc.) and does not benefit from a computerized framework. The data provided are, therefore, heterogeneous and have not been collected in the present study.

### 2.3. Statistical Analyses

Results are expressed as median (interquartile range [25–75]) for continuous variables and numbers (percentages) for non-continuous variables. When two groups were compared, we used Mann–Whitney or Fisher exact tests as appropriate. Chi-squared was used to compare frequencies when appropriate. Statistical analyses were performed using GraphPad Prism^®^, version 9.0.0.0 (GraphPad Software, San Diego, CA, USA). A *p*-value < 0.05 was considered significant.

### 2.4. Ethical Standards

The data processing of this study was carried out in compliance with the General Data Protection Regulation. This study was registered in the general register of the APHP (Treatment n° 20220408180222). The Ethical Review Committee for publications of the Cochin university Hospital (CLEP) gave a favorable opinion concerning this study: CLEP decision N°: AAA-2022-08026.

## 3. Results

### 3.1. Study Population

Between 1 January 2018 and 31 December 2021, 54,801 victims were examined at the Department of Forensic Medicine of Paris, including 2356 victims of sexual violence (4.3%). Men represented 295 victims of all victims of sexual violence (12.5%). Among them, 85 reported events that occurred before the age of 15 and were excluded (28.8%). Among the 2061 female victims, 290 reported events that occurred before the age of 15 (14.1%). Males represented 10.6% of the victims of sexual violence aged 15 years and older and 0.4% of all victims seen at the Department of Forensic Medicine during the study period. After reviewing the files, we noted that 10 patients did not report a rape or a suspected rape and were excluded. The flowchart of inclusion is presented in Figure 1.

### 3.2. Characteristics of the Victims

Age at the time of the assault

The date of the assault was specified in 199 cases. The average age of the victims at the time of the assault was of 28.8 years (median age 26.7 years [21.4–33.5]) with a maximum age of 75.8 years.

•Vulnerabilities

A vulnerability of any kind was mentioned in the files of 35 patients (17.5%):-A social or economic vulnerability linked to the reported living condition for 18 victims (9.0%),-A psychological vulnerability for 17 victims (8.5%): three victims were hospitalized in a psychiatric department at the time of the assault, four were living in a specialized center due to a mental pathology, and the files of 10 victims mentioned a psychological vulnerability (Down’s syndrome, Alzheimer’s, etc.).

•Usual treatments
-Psychiatric medications were reported by 43 patients (21.5%). Psychiatric medication intake was not associated with a suspected DFSA (*p* = 0.1568) or with alcohol or illicit substance consumption before the assault (*p* = 0.1193 and *p* = 0.2217, respectively).-Endocrine medications linked to sexual reassignment were reported by 8 patients (4.0%).-Anti-retroviral treatment not related to the assault was reported by 29 patients (14.5%), 15 in connection with positive HIV status (7.5%) and 14 with HIV pre-exposure prophylaxis (PREP, 7.0%).

•Opportunistic DFSA

A total of 128 patients (64.0%) reported voluntary use of psychoactive substances (other than medications) that facilitated the assault. Some of these patients reported the use of both alcohol and illicit substances: 28/200 (14.0%).-Alcohol use was reported by 107 patients (53.5%). The quantity was specified in 89 files. The average reported consumption was 61.8 g of pure alcohol (median 60 g [40–80]).-Illicit substance use was reported by 49 patients (24.5%). These patients reported a median of one illicit substance (excluding alcohol) [1;1]. These were mainly cannabis (16.5%), cocaine (3.0%), 3-methylmethcathinone (3MMC) (2.5%), gamma-hydroxybutyric acid (GHB) or gamma-butyrolactone (GBL) (1.5%), 3,4-methylenedioxymethamphetamine (MDMA) (1.5%), poppers (1.0%), and diverted psychoactive drugs (pregabalin 1.0%, clonazepam 0.5%). One patient reported taking methamphetamines (0.5%) and another reported alpha-pyrrolidinopentiophenone (alpha-PVP, known as “flakka”, 0.5%). Illicit substances other than cannabis were reported to have been voluntarily consumed in the context of chem-sex or slam-sex by 14 patients (7.0%).

### 3.3. Characteristics of the Assault

Location

These data were provided for 184 cases (92.0%). Most of the incidents took place at the victim’s home (23.5%) or on the public space (21.0%). Forty patients (20.0%) did not know where the suspected assault happened. Six patients described a rape during chem-sex (3.0%). Six patients described a rape in places of deprivation of liberty (3.0%).

Type

Most patients reported a rape (126/200; 63.0%), but more than one-third (74/200; 37.0%) suspected a rape but could not recall it due to amnesia of possible events.-Of the 126 patients who reported a rape, 73 (57.9%) reported a single act and 53 (42.1%) reported several acts. The acts reported were most often anal penetrations committed on the victim (108/126; 85.7%). They were most frequently penile (77/126; 65.9%) or digital (25/126; 19.8%). Six victims reported a rape with objects (6/126; 4.7%). Penile oral penetration of the victim was reported in more than one-third of cases (46/126; 36.5%). Victims less frequently reported being forced to perform penile penetration on the assailant (oral 10/126—7.9% or anal 3/126—2.4%).-For the 74 patients suspecting a rape, amnesia was mostly related to a proactive or opportunistic DFSA (62/74; 83%). Rape was suspected upon awakening, with several interrelated elements of concern: report of theft of valuables (29/74; 39%), physical injuries (24/74; 32%), patient awakening in an unusual place (22/74; 30%), or patient awakening naked in the company of another person (30/74; 41%). In ten cases, a possible assault could not be specified because of a state of psychological vulnerability of the patient (10/74; 14%; autism, psychiatric pathologies, mental retardation, etc.). For these cases, the circumstances were reported by the family or care takers. Finally, in two cases, the circumstances could not be specified because of a head injury resulting in loss of consciousness (2/74; 3%).

Associated theft

Overall, an associated theft was mentioned in 36 cases (18.0%).

Associated physical violence

Overall, associated physical violence was mentioned in 64 cases (32.0%).

Threat or use of a weapon

Four patients (2.0%) reported being threatened with a weapon (2 knives and 2 handguns). No patient reported the actual use of a weapon.

Suspicion of proactive DFSA

Almost half of the patient suspected a proactive DFSA (82/200; 41.0%). An opportunistic DFSA was statistically linked to a higher suspicion of proactive DSA (*p* < 0.0001).

### 3.4. Characteristics of the Assailant

These data were not specified in 12% of the files. Most victims identified the assailant as someone that they met on the day of the assault (49.5%). They often met on the street (18.5%), via social networks/apps (13.0%), or in a bar/nightclub (7%). The assailant was identified as a client (prostitution) in 4.0%, as a police officer in 3.0%, or as “previously unknown” without any details in 4.0%. In 24.0% of the cases, the victim suspected a rape and was unable to specify if there was any assailant. In 14.5% of the cases, the assailant was previously known by the victim: It was a friend in 8.0%, a spouse in 2.0%, and someone who shared the same living place in 4.5%.

The sex of the assailant was specified in most of the files (75.5%), and the victims almost exclusively described one or more male perpetrators (only two women were identified, sex ratio 74.5). The median number of perpetrators was of 1 [1;1], and the maximum number of 10. Multiple assailants were reported by 28 victims (14.0%).

### 3.5. Characteristics of the Forensic Examination

Time between the assault and the forensic examination

The median time was of 1 day (0–3) and the mean was of 29 days.

Extragenital findings

Somatic examination was accepted by all patients. Lesions were described in 40.0% of the cases (80/200). These were mostly bruises, abrasions, superficial wounds (93%), and, more rarely, fractured lesions (7%).

Anogenital findings

Anogenital examination was refused by 13 patients (6.5%). Anogenital lesions were described in 44 of the 187 examined patients (23.5%). These lesions were described in most cases as anal fissures, superficial tears, bruises, or superficial wounds (42/44; 96%). Two victims had rectal tears that required surgery, the assault consisted of anal penetrations, in one case by the assailant’s forearm and in the other case by a tree branch (4%).

Anal lesions were found on examination in 31 of the 83 patients reporting receptive anal penetration with a penis or object (37.0%, median time to examination of these patients: 1 day [0–3]). The presence of anal lesions was statistically higher when patients were examined within 48 h of the assault (*p* = 0.0471, compared with those examined between 48 and 96 h). Amnesia of any event was statistically related to a lower presence of anal injury (compared with patients reporting anal penetration; *p* = 0.0001).

Samples for judicial purposes

Toxicological samples were collected in 82 patients (41.0%). Genetic samples were collected from 68 patients (34.0%). The clothes worn during the assault were kept in the Department for 34 patients (34/200; 17.0%).

Management of the infectious risk

Less than half of the patients reported an initial medical consultation prior to the forensic examination, mostly in emergency rooms (40.5%). This initial consultation took place less than 48 h after the assault for 77 patients. Among these patients, 42 reported a rape requiring HIV post-exposure prophylaxis according to the current French recommendations [13]. One-third of these patients did not benefit from this prophylaxis during the initial consultation (13 patients): three were able to receive it during the forensic examination, but ten were examined after a 48 h delay. Between the initial consultation and the forensic examination, 90 patients benefited from HIV post-exposure prophylaxis (45.0%).

### 3.6. Aggravating Circumstances

Reported aggravating circumstances as detailed in French law are presented in Table 1. The total number is higher than 200, as several aggravating circumstances were found in some victims (one aggravating circumstance in median [0,1] and 0.88 in average). Apart from suspected proactive DFSA, one or more aggravating circumstances were found in the files of 81 victims (40.5%). Including suspected proactive DFSA, one or more aggravating circumstances were found in the files of 141 victims (70.5%). To note, opportunistic DFSA is not an aggravating circumstance under French Law.

### 3.7. Characteristics of the Victims and the Assaults According to the Victim’s Vulnerability

We compared the characteristics of the victims and the assaults, according to the victim’s vulnerability; the results are presented in Table 2.

### 3.8. Characteristics of the Victims and the Assaults According to the Victim’s Memory of the Assault

We compared the characteristics of the victims and the assaults, according to the victim’s memory of the assault; the results are presented in Table 3.

### 3.9. Characteristics of the Victims and the Assaults According to the Assailant’s Status

We compared the characteristics of the victims and the assaults, according to the assailant’s status; the results are presented in Table 4.

Victims assaulted by a previously known individual were more likely to be vulnerable or to take psychiatric medications. They reported less frequently an initial medical consultation and presented fewer injuries.

## 4. Discussion

According to data available in the international literature, men are less affected by sexual violence than women. Our study confirms this observation: Over a four-year period, between 2018 and 2021, in the Department of Forensic Medicine of Paris, 12.5% of all victims of sexual violence were male. Sexual violence is also thought to be particularly under-reported by men [14]. Males represented 10.6% of all victims of rape or suspected rape, aged 15 years of age and older. In France, according to the report of the survey “Cadre de vie et sécurité” (living conditions and security), 16.1% of the adults (18–75 years old) that reported a rape or an attempted rape were men [4]. Our results confirm the under-reporting of sexual violence among men. Male victims of rape or suspected rape were mostly young, with an average age of 28.8 years. These results are consistent with the existing literature in France: the mean ages were 25 and 29.5 years in two similar studies in Bordeaux and Bondy, France, respectively [15,16]. Most of the patients reported anal penetration, committed by a single male assailant, whom they met on the day of the assault, often in a festive context. Victims more rarely reported being forced to commit penile penetration on the assailant. Since the reform of August 03, 2018, changes in French legislation allow for the circumstance of having been forced to commit a sexual act on the perpetrator to be taken into account [17]. This modification allowed the filing of a complaint for 13 patients in our study.

Most patients were examined shortly after the assault (median delay of 1 day). An early examination allows for better detection of possible lesions, especially in the anal mucosa, which heals rapidly. For superficial lesions such as peri-anal abrasions, the reconstruction of the epithelium can be complete within 24 h in children, thus masking the lesions [18]. Healing of superficial lesions is generally effective within a few days. In our study, anal lesions were found on examination in more than one-third of patients reporting anal penetration (penis or object) regardless of the time frame. However, the presence of anal lesions was statistically higher when patients were examined within 48 h of the assault. The presence of anogenital injuries does not provide a diagnosis of rape. These injuries are not specific to traumatic intercourse and can be found after consensual intercourse in both males and females [19]. Nevertheless, the identification of anogenital or general lesions is generally associated with a higher rate of conviction of the aggressor [20]. In the literature, the prevalence of anogenital injuries in male victims of sexual assault varies between 15% and 61.4% [19]. A timely examination also allows for better medical management, as the initiation of a HIV post-exposure prophylaxis is most effective within 48 h after the event, according to the current French recommendations [13]. In our study, ten patients could have benefited from a HIV post-exposure prophylaxis in the emergency room during the initial examination. This finding illustrates the lack of information among emergency physicians regarding the management of rape victims.

Among the patients in our study, 17.5% were vulnerable (psychologically, socially, or economically). As previously described in the literature, vulnerable patients were more likely to have been assaulted by someone they knew [21]. They were less likely to report an associated suspicion of proactive DFSA, their vulnerability having itself facilitated the crime of the aggressor.

From other studies, psychological vulnerability seems to be associated with a higher risk of sexual violence. A study in Sweden showed that diagnosis of psychotic, bipolar, stress anxiety disorders, and depression were more common among patients with a diagnosis of sexual abuse [22]. A study carried out in Seine-Saint-Denis, France, showed that male victims were more frequently disabled or vulnerable than were female victims, and that vulnerable male victims reported previous sexual assaults more frequently than other male victims [15]. More than one-fifth of patients in our study also reported the use of psychiatric medication (antidepressants, anxiolytics, etc.). France is a country with a high consumption of antidepressants and anxiolytics; a report by the French National Agency for the Safety of Medicines and Health Products (ANSM) estimated that 13.4% of the French adult population had taken a benzodiazepine at least once during the year of 2015 [23]. The consumption of psychiatric drugs is also on the rise, particularly in connection with the COVID-19 pandemic [24]. In our study, vulnerable patients were more likely to take psychiatric medication, which is consistent with psychological vulnerability. Psychiatric medication intake was not associated with a higher consumption of alcohol or illicit drug on the day of the assault.

Vulnerability related to living conditions is not traditionally investigated in studies of male victims of sexual violence. In France, social and economic vulnerability was recently added to the aggravating circumstances of sexual violence by the law as of 3 August 2018 [17]. In our study, nearly one-tenth of the victims reported a social or economic vulnerability linked to their living condition. Being homeless has been proven to be associated with a high risk of rape in the United State [25,26]. Sexual violence may be particularly under-reported by this population, especially as homeless people are afraid of being discriminated against by the police [27]. If social workers are aware of the link between homelessness and sexual violence, preventative actions on this theme should be carried out with police officers.

More than sixty percent of the patients of our study reported the voluntary consumption of alcohol and/or illicit substances prior to the assault, characterizing a potential opportunistic DFSA. Opportunistic DFSA is often confused with proactive DFSA, in which the administration of the psychoactive substance is done by the assailant without the victim’s knowledge in order to facilitate a sexual assault [28]. In line with the international literature, the report of an opportunistic DFSA was associated with a higher suspicion of proactive DFSA in our study [29,30,31,32,33]. In other studies, in France and in the UK, a recent psychoactive substance use (including alcohol) was reported by, respectively, 26% and 58% of male victims of a sexual assault [8,15]. Our study found a higher proportion of recent psychoactive substance use (including alcohol). This finding may be related to the more festive Parisian population, as well as to the significant proportion of assaults that took place in a chem-sex or a slam-sex context. It was not possible to characterize the actual proportion of proactive DFSA, as the samples for toxicological purposes are analyzed in expert laboratories under the responsibility of the judicial authority.

More than half of the patients reported the consumption of alcohol (53.5%). The median alcohol consumption was over 60 g of pure alcohol, three times greater than the recommended amount as a daily maximum in France [9]. Ethanol will decrease the vigilance of victims who will be less able to identify the administration of a substance without their knowledge. It can also potentiate the effects of other psychoactive substances. Alcohol consumption, in the form of alcohol use disorder (AUD) or binge drinking, has been proved to be associated with sexual violence [34]. In another French study, 22% of the male victims of sexual violence reported an alcohol consumption before the assault [15]. AUD was not explored in our study, but in other cohorts of male victims of sexual assault, addictive disorders were reported by 22% of the patients in France and 19.2% in Italy [15,20]. These results underline the importance of prevention strategies and information on the risks associated with ethanol consumption, especially for a young and festive population.

The use of illicit substance(s) use was reported in one-quarter of the patients. The primary substance used was cannabis, which is the most-consumed illicit substance in France [35]. France is also the European country where cannabis consumption is the most prevalent among young adults [35]. The intake of cannabinoids can induce neuropsychic disorders (euphoria, imaginative exaltation with impaired judgment, and potential drowsiness) and variable sensory manifestations that can go as far as hallucinations or cognitive disturbances [36]. This symptomatology can, therefore, induce an opportunistic DFSA. Almost one-tenth of the patients reported the use of other illicit substances in the context of chem-sex or slam-sex. Chem-sex is defined as the absorption of psychoactive substances during sexual intercourse, which can be swallowed, snorted, or inserted rectally. Slam-sex is a variant where the substances are injected [37]. These practices are carried out mainly by men who have sex with men (MSM). The main substances used are cathinone, methamphetamine, GHB/GBL, or ketamine. These sexual practices are considered particularly risky. The combination of several substances is frequent, increasing the risk of potentially lethal overdose and of substance use disorder. The risk of infection is also often less controlled during these practices [37]. Our study underlines the fact that, in addition to the toxic and infectious risks, chem-sex can lead to sexual violence. There is likely a significant under-reporting of this type of incident. These victims may fear that they will not be believed by the judicial authorities, or even that they will be prosecuted for the use of illicit substances. Most of the substances involved in chem-sex are indeed under international control and scheduled under the Convention of Psychotropic Substances of 1971. In practice, in France, patients filing a complaint in a chem-sex context are not prosecuted for substance use. The question of reception at the police station and the establishment of absence of consent is more complex. With the use of psychoactive substances, it is indeed sometimes complex to identify what the victim may have consented to, as well as what was understood by the assailant(s). Nevertheless, not respecting a change of mind or a limit established by a partner, regardless of his or her state of impregnation with a substance, is constitutive of sexual violence. People practicing chem-sex should be more informed about the particularity of consent in this context. It also seems essential to reinforce prevention and information actions, to help them assert their rights in cases of sexual violence. Informational materials could be shared through social networks, specific associations, or even dating applications.

In our study, more than one-third of the patients suspected a rape but could not recall it. These patients had more frequently consumed alcohol and/or illicit drugs, but not necessarily in a chem-sex context. They were more likely to suspect a proactive DFSA and reported more frequently an associated theft. Although there was no difference in terms of delays of examination, fewer anogenital lesions were found in these patients. Total amnesia can often be explained by alcohol intoxication and/or proactive DFSA. In most studies, total amnesia is reported by less than 10% of the victims [38,39,40]. The large percentage of victims who reported alcohol and/or illicit substances use might explain the higher prevalence in our study.

Finally, our study explored the existence of aggravating circumstances. These could be related to the victim (vulnerability) or the circumstances of the aggression. We noted only one perpetrator in the family context and only three marital rapes. The under-reporting of sexual violence perpetrated by a known perpetrator is a well-described phenomenon, which our work confirms [41]. Unknown perpetrators are more likely to be reported, as there is less fear of reprisals or of upsetting the family environment. This phenomenon is described similarly in the two other French studies on the subject, with most perpetrators being male and unknown in 38.4% of cases in Bordeaux and 39.8% of cases in Seine-Saint-Denis. Most assaults occurred in the victim’s home (23.1%) or on the public space (21%). According to some authors, assaults on the public space are more frequent among homosexual men, in connection with sexual encounters in parks [16]. Although nearly 20% of the victims reported having met the perpetrator on the public space and 13% via dating applications, this hypothesis was not explored in our study, as sexual orientation is not discussed during the examination. Our study also found six rapes during deprivation of liberty. All these patients reported digital anal penetration during body searches by police officers. These acts were probably more akin to humiliation than to sexual gratification for the perpetrators. Acts of rape in prison or police custody are well-described in the literature [1]. The absence of reported rapes in prisons in our study is likely related to its geographical specificity as there is only one prison in Paris, the Santé prison. It was closed in 2018 and housed a maximum of 920 inmates, which, combined with a probable under-reporting of rapes in prisons, may explain the absence of cases in our study. Among the aggravating circumstances, our study found 7% of prostitutes, often transgender or transsexual. Sex workers represent a category of patients particularly exposed to sexual violence. The 2016 law, aimed at reinforcing the fight against the prostitution system and supporting prostitutes, abolished the offense of passive soliciting and prohibited the purchase of sexual acts by clients [42]. This law, intended to protect prostitutes, ultimately exposes them to more sexual violence. Indeed, to maintain a clientele, prostitutes will tend to carry out their activities in more isolated places and will, thus, be particularly vulnerable. It seems essential to reinforce the protection of prostitutes and their support in case of sexual violence. Overall, apart from suspected proactive DFSA, we found a high proportion of aggravating circumstances (40.5%). Including suspected proactive DFSA, one or more aggravating circumstances were found in the files of 141 victims (70.5%).

Our study presents the characteristics of male victims of rape or suspected rape, examined following a complaint in Paris between 2018 and 2021. It has certain limitations, inherent to its retrospective and mono-centric methodology. There is a significant risk of under-reporting of certain data, notably on alcohol or illicit substances consumption. Our sample size is also limited, and the Parisian population may have certain specificities, making the results difficult to extrapolate to a rural population, for example. Finally, the forensic Department of Paris receives patients only after a formal report to the police which. This may explain the high percentage of associated aggravating circumstances, as victims often feel that their chances of being believed increase when the facts are perceived as more serious. This factor could also limit the representativeness of the results. Finally, it would be interesting to confront the results of our study to the judicial follow-up of the investigations, notably to establish whether the presence of injury leads more often to a conviction of the assailant.

## 5. Conclusions

The data available on sexual violence reflect only a tiny part of the phenomenon [1]. The underestimation of rape victims is significant, especially among men. This subject is still taboo in our society even if sexual violence has a profound impact on physical and mental health. Our study provides epidemiological data and highlights the significant proportion of victims with psychological, economic, or social vulnerabilities. We also highlight the important proportion of sexual violence or suspicion of sexual violence occurring in a festive context, among victims who have consumed alcohol, and/or illicit substance, sometimes in a chem-sex context. Preventative actions aimed at men need to be implemented, especially for those who practice chem-sex or binge-drinking. These actions should aim not only to relieve the victims of their guilt and improve their medical and psychological care, but also to ensure that they report sexual violence to the police, whatever the context.

## Figures and Tables

**Figure 1 ijerph-19-13909-f001:**
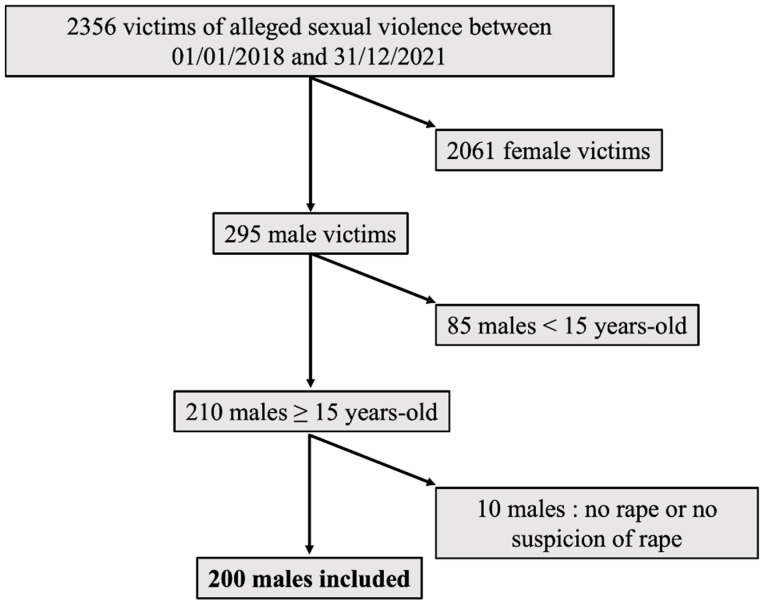
Flow chart of inclusion.

**Table 1 ijerph-19-13909-t001:** Number of victims describing aggravating circumstances.

Aggravating Circumstance	Number of Victims (%)
Act resulting in permanent mutilation or disability	2 (1.0)
Act committed on a minor of 15 years old	not included in the study
Physical/psychological vulnerability of the victim	17 (8.5)
Economic/social vulnerability of the victim	18 (9.0)
Act committed by an ascendant/person in authority	1 (0.5)
Act committed by a person who abuses authority *	6 (3.0)
Act committed by several persons	28 (14.0)
Act committed with the use or threat of a weapon	4 (2.0)
Contact with the perpetrator through (…) an electronic communication network	not included in the study
Act committed in conjunction with one or more other rapes committed on other victims	0 (0.0)
Act committed by a spouse	4 (2.0)
Act committed by a person acting in a state of obvious intoxication or under the influence of drugs	not included in the study
Act committed on a person engaged in prostitution	14 (7.0)
Minor present at time of offence	0 (0.0)
Proactive DFSA	Suspicion: 82 (41.0) **

* This category includes patients reporting rape during a deprivation of liberty measure at the time of the assault (police custody or administrative detention center). In all cases, it was a case of digital anal penetration during the search, committed by police officers. ** Some of the patients (*n* = 62) did not remember the eventual facts and, therefore, suspected DFSA. Another portion (*n* = 20) reported a rape, which they thought had been facilitated by drugs.

**Table 2 ijerph-19-13909-t002:** Characteristics of the victim and the assault according to the victim’s vulnerability. The bold number were for all *p* < 0.005 (significant).

	Vulnerable *n* = 32 Number (%)	Non Vulnerable *n* = 168 Number (%)	*p*
**Victim**			
Median age (years)	25.93	26.98	0.5280
Psychiatric medication	17 (55)	35 (20.8)	**0.0001**
Alcohol use before the assault	9 (29)	70 (41.7)	0.1509
Psychoactive substance use before the assault	12 (38)	37 (22.0)	0.0621
Initial medical examination	13 (41)	65 (38.7)	0.9371
Median delay between the assault and the forensic examination (days)	1 day	1 day	0.9110
Extragenital lesions	9 (28)	71 (42.3)	0.1346
Anogenital lesions	3 (9)	41 (24.4)	**0.0381**
**Assault**			
Number of assailants (mean)	1 (3)	1 (0.6)	0.3199
Previously known assailant	13 (41)	18 (10.7)	**0.0001**
Place of assault			
Assailant’s home	3 (9)	28 (16.7)	0.3242
Victim’s home/place of living	15 (47)	30 (17.9)	0.0002
Public place	4 (13)	38 (22.6)	0.2232
Type of assault			
Cannot recall any assault	13 (41)	61 (36.3)	0.6431
Anal penetration of the victim	16 (50)	82 (48.8)	0.9017
Oral penetration of the victim	8 (25)	39 (23.2)	0.8272
Penetration of the assailant	22 (69)	11 (6.5)	0.9501
Physical violence	6 (19)	58 (34.5)	0.0796
Theft	1 (3)	35 (20.8)	**0.0169**
Suspicion of proactive DFSA	5 (16)	77 (45.8)	**0.0015**

**Table 3 ijerph-19-13909-t003:** Characteristics of the victim and the assault according to the victim’s memory of the assault. The bold number were for all *p* < 0.005 (significant).

		Patient Reporting a Rape *n* = 126 Number (%)	Total Amnesia *n* = 74 Number (%)	*p*
**Victim**				
Median age (years)		24.67	28.52	**0.0045**
Psychiatric medication		33 (26.2)	19 (26)	0.9361
Vulnerability		19 (15.1)	13 (18)	0.6431
Alcohol use before the assault		49 (38.9)	58 (78)	**<0.0001**
Psychoactive substance use before the assault		17 (13.5)	32 (43)	**<0.0001**
Initial medical examination		50 (39.7)	28 (38)	0.7962
Median delay between the assault and the forensic examination (days)		1 day	1 day	0.8205
Extragenital lesions		54 (42.9)	26 (35)	0.2818
Anogenital lesions		34 (27.0)	6 (8)	**0.0001**
Chem-sex		6 (4.8)	8 (11)	0.1055
**Assault**				
Number of assailants (mean)		1.25	1.8	**0.0049**
Previously known assailant		23 (18.3)	8 (11)	0.1602
Place of the assault				
	Assailant’s home	22 (17.5)	9 (12)	0.3175
	Victim’s home/place of living	31 (24.6)	14 (19)	0.3527
	Public place	36 (28.6)	6 (8)	**0.0006**
	Unknown place	0 (0.0)	35 (47)	**<0.0001**
Physical violence		64 (50.8)	0 (0)	**<0.0001**
Theft		7 (5.6)	29 (39)	**<0.0001**
Suspicion of proactive DFSA		20 (15.9)	62 (84)	**<0.0001**

**Table 4 ijerph-19-13909-t004:** Characteristics of the victim and the assault according to the assailant’s status. The bold number were for all *p* < 0.005 (significant).

		Known Assailant *n* = 31 Number (%)	Previously Unknown Assailant *n* = 96 Number (%)	*p*
**Victim**				
Median age (years)		26.7	25.5	0.8699
Psychiatric medication		21 (68)	12 (13)	**<0.0001**
Vulnerability		13 (42)	11 (11)	**0.0002**
Alcohol use before the assault		12 (39)	45 (47)	0.4268
Psychoactive substance use before the assault		11 (35)	21 (22)	0.1292
Initial medical examination		9 (29)	43 (45)	**0.0411**
Median delay between the assault and the forensic examination (days)		1 day	1 day	0.2945
Extragenital lesions		7 (23)	42 (44)	**0.0055**
Anogenital lesions		1 (3)	35 (36)	**0.0004**
Chem-sex		0 (0)	9 (9)	0.0770
**Assault**				
Number of assailants (mean)		1.2	1.4	0.7958
Place of the assault				
	Assailant’s home	5 (16)	24 (25)	0.3063
	Victim’s home/place of living	20 (65)	17 (18)	**<0.0001**
	Public place	1 (3)	30 (31)	**0.0016**
Physical violence		7 (23)	47 (49)	**0.0098**
Theft		0 (0)	13 (14)	**0.0306**
Suspicion of proactive DFSA		6 (19)	32 (33)	0.1395

## Data Availability

Data supporting results can be requested to laurene.dufayet@aphp.fr.

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
