# Peer review of "Male Victims of Rape: An Observational Study over Four Years in Paris, France"

_ijerph, 2022, doi:10.3390/ijerph192113909_

Round 1

Reviewer 1 Report

Congrats for the work done, very intresting.

I have only few observation:

- keywords: add vunerability

- Materials and Methods 2.1: Streamline the paragraph

- Tables: If you put (%) in the top of the table, there is no reason to repeat it in all subsequent items

- Discussion: Streamline the paragraph

Author Response

We thank the reviewer for her/his favorable opinion and valuable comments. 

Congrats for the work done, very intresting.

I have only few observation:

- keywords: add vunerability : this has been added 

- Materials and Methods 2.1: Streamline the paragraph : we tried to make this paragraph more readable

- Tables: If you put (%) in the top of the table, there is no reason to repeat it in all subsequent items : this was modified 

- Discussion: Streamline the paragraph :  we tried to make this paragraph more readable

Reviewer 2 Report

This is a well-written paper and it touches on a very important but currently neglected topic.

Forensic data on male victims of male-to-male sexual violence/attack/rape provides clear evidence for future policy, program and health promotion. 

My only suggestion would be to ask authors consider adding something about the self-reported nature of the cases in the administrative forensic data, acknowledging it's probably a minority that are visible; and further discuss programs to raise awareness in social and digital scenes among gay and other men who have sex with men seeking potential male-to-male sex.

Author Response

We thank the reviewer for her/his favorable opinion and valuable comments. 

This is a well-written paper and it touches on a very important but currently neglected topic.

Forensic data on male victims of male-to-male sexual violence/attack/rape provides clear evidence for future policy, program and health promotion. 

My only suggestion would be to ask authors consider adding something about the self-reported nature of the cases in the administrative forensic data (this was added), acknowledging it's probably a minority that are visible; and further discuss programs to raise awareness in social and digital scenes among gay and other men who have sex with men seeking potential male-to-male sex. the discussion was streamlined according to the other reviewer's comment, we have tried to improve the discussion about men who have sex with men.